# Insight into the Recent Application of Chemometrics in Quality Analysis and Characterization of Bee Honey during Processing and Storage

**DOI:** 10.3390/foods12030473

**Published:** 2023-01-19

**Authors:** Maria Tarapoulouzi, Monica Mironescu, Chryssoula Drouza, Ion Dan Mironescu, Sofia Agriopoulou

**Affiliations:** 1Department of Chemistry, Faculty of Pure and Applied Science, University of Cyprus, P.O. Box 20537, Nicosia 1678, Cyprus; 2Faculty of Agricultural Sciences Food Industry and Environmental Protection, Lucian Blaga University of Sibiu, Bv. Victoriei 10, 550024 Sibiu, Romania; 3Department of Agricultural Production, Biotechnology and Food Science, Cyprus University of Technology, P.O. Box 50329, Limassol 3036, Cyprus; 4Department of Food Science and Technology, University of the Peloponnese, Antikalamos, 24100 Kalamata, Greece

**Keywords:** honey, quality parameters, chemometrics, storage, processing, thermal treatment

## Abstract

The application of chemometrics, a widely used science in food studies (and not only food studies) has begun to increase in importance with chemometrics being a very powerful tool in analyzing large numbers of results. In the case of honey, chemometrics is usually used for assessing honey authenticity and quality control, combined with well-established analytical methods. Research related to investigation of the quality changes in honey due to modifications after processing and storage is rare, with a visibly increasing tendency in the last decade (and concentrated on investigating novel methods to preserve the honey quality, such as ultrasound or high-pressure treatment). This review presents the evolution in the last few years in using chemometrics in analyzing honey quality during processing and storage. The advantages of using chemometrics in assessing honey quality during storage and processing are presented, together with the main characteristics of some well-known chemometric methods. Chemometrics prove to be a successful tool to differentiate honey samples based on changes of characteristics during storage and processing.

## 1. Introduction

Bee honey is a natural food extracted from honeycombs, together with bee wax, pollen and propolis, royal jelly and venom [1,2]. Honey, also known as “the food of the Gods” [3], was always used as food by humans [4]; its success is represented by the very high content in simple sugars (honey is a supersaturated sugar solution with about 25–35% glucose and 35–45% fructose, together with 1–2% sucrose) [5]. Honey is considered to be the most energy-dense food in nature [6], with an appreciable energy value (100 g of bee honey provides 310 kcal) [7]. It has a glycemic index in the range 32–87 [8], seeming to show a hypoglycemic effect [9].

Honey is a complex food, a valuable gift of nature [10,11]; research in the field shows that, in addition to the sugars, bee honey also contains valuable compounds in small quantities, namely: vitamins (B1, B2, B6, C, K, A, D, E, K, etc.), proteins (an average of 0.5% in the case of floral honey and higher doses in the composition of manna honey), dextrins, coloring and odorous substances, organic acids (malic, pantothenic, citric, oxalic, lactic, succinic, etc.), trace elements (calcium, iron, potassium, silver, nickel, beryllium, etc.) and others [9,12,13]. Because of this very complex composition and valuable compound richness, bee honey is used in its natural state [14,15] or incorporated in food matrices to improve sensorial [16], functional and physicochemical properties [17]. The antimicrobial [18,19], antioxidant [20,21,22], antiviral [23] and anti-inflammatory action [24] of different types of honey is well known; the research has also demonstrated the anticarcinogenic activity of honey [25].

The addition of food ingredients is prohibited in honey, including food additives, as well as any other addition. Similarly, the removal of any of the natural components of honey is prohibited, including pollen, unless such removal cannot be avoided [26]. The authenticity of honey can be defined as the precise identification of its botanical and geographic origin in relation to its unique composition and properties [27]. Honey adulteration has generally been identified with trained sensory evaluation based on viscosity, color, taste and aroma The testers in such sensory evaluations require long-term professional training to ensure the reliability of the results. However, these methods are mostly subjective and prone to human error. Beyond the sensory assessment, methods described in the literature include microscopy, immunoassays, physico-chemical analyses, chromatography, mass spectrometry, near-infrared spectroscopy, Raman spectroscopy, DNA transcoding and enzyme-linked immunosorbent assay [28].

In general, the quality of honey and its chemical composition are related to many factors, such as its geographic, botanical or plant origin, climate and seasonality; other factors could be external, such as environmental factors, processing methods of honey by beekeepers, storage conditions and deliberate adulteration by producers [29]. The quality of honey can also be characterized by its purity; the purity of honey can be determined by its physico-chemical properties, i.e., moisture content, pH, free acidity, total soluble solids, sugar content, color intensity, 5-hydroxymethylfurfural (5-HMF) content and amino acid content [30]. Heavy metals can accumulate in the honey, from the soil [31] or from the air [32]. Therefore, the evaluation of physico-chemical properties and certification of botanical origin is vital to determine the quality and authenticity of honey. These factors may affect market price and consumer acceptance. Consumers’ safety and their protection from fraud is the overall goal [33].

For production on an industrial scale, raw bee honey has to be extracted from the honeycombs and then processed. Due to their natural characteristics, all varieties of flower honey, except acacia honey, crystallize shortly after extraction from honeycombs; crystallization of flower honey is a natural process and is due to the fact that most varieties of honey are supersaturated solutions of sugars (at an average temperature of 20 °C) [34]. The honey crystallization degree is dependent on the absolute pollen count [35]. Crystallization does not alter the biological and medicinal properties of the product and should not be considered as a defect, but rather as a guarantee of its authenticity [36]. However, many consumers prefer fluid honey and have reservations about buying and consuming crystallized honey [37]. In order to meet the demands of consumers, many solutions are used to obtain decrystallized honey and to keep it in this state for as long as possible; on the other hand, it is very important for the technologists to bring honey in a liquid state through liquefaction for further processing [34]. After liquefaction, technologies such as ultrasonication, microwave and infrared irradiation or heating are used to eliminate water in excess and to sterilize the raw bee honey, to continue honey decrystallisation and to temper it for further processing steps such as filtration or mixing with other compounds [38].

Many research papers and reviews indicate the difficulties in analyzing the great amount of results generated by the literature, but also the disparity of the results, due to different methods used to appreciate honey quality [9,19,22,39,40,41,42]. The progress of methods and techniques used, the multivariate methods applied to analyze honey, indicates chemometrics as one of the most adequate solutions for large quantities of data evaluation and interpretation [43]. Nowadays, the field of chemometrics is well-applied in the fields of honey authenticity and quality control. There are numerous studies published in the last few years, e.g., between 2017 and 2021 combining chemometrics with well-established analytical methods in honey authentication and quality control [44,45,46,47,48,49,50,51,52,53,54,55,56,57,58,59,60,61,62,63,64]. Based on chemometrics, Bruker has developed the largest database containing the spectroscopic ^1^H NMR signatures of 28,000 reference honeys, monofloral and polyfloral types from more than 50 countries, aiming at the botanical and geographical determination or adulteration of honeys [65]. This review aims to present the state-of-the-art in using chemometrics in analyzing honey quality during processing and storage; the advantages of using chemometrics are presented extensively in the next sections, and also the main characteristics of some well-known chemometric methods. From our knowledge, no review on the chemometric methods used for assessing honey quality during processing and storage has been published and this is the most original contribution of this publication.

## 2. Methodology and Design

The literature search was conducted in order to identify the most relevant research articles and reviews illustrating the state-of-the art in using chemometrics in analyzing bee honey quality during processing and storage. The online databases queried were Web of Science, ScienceDirect, Google scholar, MDPI, ResearchGate, PubMed, Scopus and the Wiley Online Library. The keywords used individually and combined on the searching engines were: honey, quality parameters, chemometrics, storage, processing, thermal treatment. By selecting the publications to be included, a similar pathway as that in [66] was used. The criteria were: (i) research that focused on the use of chemometric methods in analyzing honey quality and changes/adulteration during storage and processing; (ii) papers written in English; (iii) papers with accessible full text (in some cases after requests from the authors). 

After a short introductory discussion on honey as food and its properties, we discuss the aspects related to storage and the two main processing steps—dewatering and thermal treatment—and their influence on the honey quality (Section 3). Then, we present the main chemometric methods and the quality parameters of honey mostly investigated by using chemometrics and we intensively analyze the main publications applying chemometrics in honey processing and storage (Section 4).

The review includes mainly publications from the period 2010–2022 (the last 12 years), together with a few older publications considered as being very important for this research.

## 3. Honey Processing and Storage

After the extraction from the honeycomb, honey contains pollen, beeswax and other materials, and these impurities have to be removed from the preheated honey by straining and coarse filtration. Dewatering, liquefaction and filtration are the main operations in processing honey [67] to assure its commercial quality and these processes will be discussed here, together with storage before or after processing. The product can be bottled as it is or can be enriched with other different ingredients, including: (a) bioactive compounds such as propolis or bee bread [68,69], coumarin [70] or spirulina [71]; (b) dried fruits [72,73,74]; (c) flavored materials usually accepted by the consumers, such as cocoa or cinnamon [75,76], to obtain novel foods. Figure 1 presents the main industrial treatments of the raw honey received from the primary suppliers and/or beekeepers.

According to the EU Regulations, the compositional criteria of honey are: moisture content (M), simple sugars (fructose + glucose and sucrose), water-insoluble solids, electrical conductivity, free acids, 5-hydroxymethylfurfural (HMF) content and diastase value (the last two criteria in the processed honey) [39,77] together with melissopalynological analysis and other physical–chemical parameters (such as pH, proline content, enzymatic activity and ash content) and with sensorial characteristics (such as color and volatile profile), they are the classical tools to provide important information on the honey’s origin and authenticity [78]. Other important parameters, especially for honey storage and processing, are: water activity (a_w_) and rheological behavior [79,80]. HMF and M can be used to classify blended honey samples according to their crystallization degrees [35]. According to the European and international norms, a special importance must be given to the knowledge of these quality parameters, but also to the causes that determine changes in their initial values [36,81,82]. 

### 3.1. Influence of Storage Conditions on Raw Honey Quality

After the extraction from the comb, bee honey could be preserved for a long time (from months to years). The storage conditions will influence honey quality; refrigeration temperatures lead to a darkening of the color, whereas storage at room temperatures and higher (28 °C) results in modification of the acidity and an increase in HMF and the microbiota in honeys, together with a small modification of the composition in monosaccharides [83] and phenols [84]. Although low temperatures do not favor honey crystallization, its refrigeration determines the internal reorganization of the water molecules with increased fermentation risk when the honey returns to room temperature [85].

There is a negative correlation between HMF content and diastase (amylase) activity during storage; a longer storage time increases the HMF content, with a reduction in the enzymatic activity [83]. The storage of honey at higher temperatures (35 °C) for a longer time (from 6 to 9 months) increases the HMF content to values exceeding those accepted by the legislation [86]. A deeper discussion on these parameters is given in Section 4.2. of this review.

The storage conditions influence the antibacterial properties of honey. Besides pro-polis, two antibacterial compounds are the most investigated in honey, methylglyoxal, the compound responsible for the antimicrobial action of Manuka honey [87] and hydrogen peroxide obtained from glucose degradation by glucose oxidase in bee honey [88,89]. It should be mentioned that the peroxidase activity of honey is lost during storage, together with the exposure to light and heating [90,91,92].

The aroma compounds in honey are a complex mix of volatile components with various functions and relatively low molecular weight [93]. The storage results in reduction in its odorous compounds, especially at higher temperatures; temperatures near to refri-geration maintain the volatile compounds in honey [94]. 

The honey properties during storage seem to be more dependent on the honey type than the storage conditions [95].

### 3.2. Influence of Storage Conditions on Raw Honey Quality

Dewatering and dehumidification refer both to the Moisture (M); M can be removed from honey through dewatering or through dehumidification from air in contact with honey. The acceptable range of moisture content is 16.4–20.0% [96]; higher moisture can cause honey fermentation by osmotolerant yeasts [97] and favors faster crystallization [98]. Table 1 presents some methods used for M removal. 

### 3.3. Honey Thermal Treatment

The thermal treatment of honey involves two main operations, liquefaction and pasteurization. Liquefaction is realized for the honey decrystallization, by using different methods. The classical liquefaction is conducted by heating honey at temperatures around 50 °C (45 to maximum 60 °C) for a long time (12 h), with sensorial changes such as: darkening, contracting the taste of caramel and weakening or even the disappearance of the specific flavor [106]; the antioxidant activity and the phenolic compounds seem to also become modified [106,107]. 

Another method used is microwave heating, with the increase in HMF concentration and decrease in antioxidant properties as disadvantages [108]. Novel methods such as ultrasounds or high-pressure treatments seem to be effective in destroying/minimizing the crystals or in increasing the period before starting crystallization and they have the advantage of being shorter [109,110,111]. Higher ultrasound power input lowers the liquefaction time (in minutes) [112]. Ultrasounds are beneficial for the extraction of volatile and semi-volatile compounds [113], and they do not affect the physical–chemical parameters (moisture, pH, diastase activity, HMF content) and sensorial characteristics (color) [114,115]; ultrasounds have antimicrobial activity, too, depending on the pathogens and honey types [116,117,118]. Studies have shown that the high-pressure treatment improves the nutritional and antimicrobial characteristics of honey [110,119].

Liquefied honey can be pasteurized at higher temperatures (around 80–90 °C) for seconds to destroy microorganisms (especially yeasts, responsible for fermentation); a se-condary effect is the moisture reduction and the delaying of crystallization [120]. The non-thermal treatments such as ultrasounds [121] or irradiation [122] are promising techniques for destroying the microorganisms in honey. In addition to the classical way of heating the water for obtaining the temperatures required for the thermal treatment (liquefaction or pasteurization), a new trend is to use geothermal water as a sustainable resource [123,124]. 

## 4. Chemometrics Used in Honey Quality Analysis during Storage and Processing

### 4.1. Introduction to Chemometrics in Honey Quality Analysis

Chemometrics means the use of statistical and in recent times Artificial Intelligence (AI) methods to characterize and classify samples based on large quantities of analytical data. The most frequent applications are the identification of the class of a sample and the prediction of its properties not covered by the analysis [125]. The methods can be divided into unsupervised and supervised ones. 

#### 4.1.1. Unsupervised Chemometric Methods

The unsupervised methods do not require a training set, and they group the samples based on the similarities between them; a new sample will be placed in the existing groups or possibly will create a new group. These methods are suited for preliminary, exploratory and qualitative analysis especially when no prior classifying information is available. Examples of such methods are: ANOVA or ANalysis Of VAriance is used to compare statistical populations in order to decide if there are statistically significant differences between them. Its use has become a standard requirement for proving the soundness and validity of a research hypothesis. In the context of chemometrics, ANOVA is used to investigate the effect of independent variables on the dependent variable. If multiple dependent variables are of interest then a Multivariate ANOVA (MANOVA) is performed [126];Cluster Analysis (CA) which groups samples in clusters with the most used being:
oHCA or Hierarchical Cluster Analysis [127], which uses distance-based methods to group the data in hierarchical clusters and to place a new sample in this hierarchy;oK-means clustering, which is a non-hierarchical clustering of data in k clusters. 


Examples of the use of clustering techniques in honey chemometrics can be found in [128,129]. In these individual pieces of research, polycyclic aromatic hydrocarbons (PAHs) were identified in honey by using ultrasound-vortex-assisted dispersive liquid–liquid micro-extraction followed by a triple quadrupole gas chromatograph/mass spectrometer (DLLME-GC-MS). The dataset was then grouped using k-Mean cluster analysis and PCA in order to identify geographic specific pollution [128] or human activity specific pollution patterns [129].

Principal Component Analysis (PCA) is used to reduce the dimensionality of a sample space when many features are investigated for many samples; they are plotted in a reduced space where the axes are combinations of the features chosen so that the relations between them (distances) are preserved [130]. PCA principal use is for visualization and qualitative analysis and it needs the use of a secondary method—usually a supervised Discriminant Analysis (DA) method—for classification. In [128,129], PCA is used in combination with k-Mean cluster analysis to visualize the grouping of pollutants based on geographical location [127] or human activity [128]. PCA was used by [131] to cluster honey types based on the data expressing the content in vi-tamin B2 and Cu and the antioxidant activity measured by 2,2′-azino-bis(3-ethylbenzothiazoline-6-sulfonic acid (ABTS) [132] and CUPric Reducing Antioxidant Capacity (CUPRAC) [133] values. The grouping allows the identification of the botanical origin:
oAn extension to multiple dimensions of PCA is the PARAllel FACtor analysis (PARAFAC) [134,135,136] which can be used on multiway spectral data. It is employed in [134] where fluorescence spectrometry data are first decomposed with PARAFAC in order to identify the representative patterns in honey. An improvement of the traditional PARAFAC specifically for use on chromatography data is the alternating trilinear decomposition algorithm (ATLD) [137]. ATLD can be used to decompose the HPLC data in order to evidence the data related to the phenolic components used as markers; the quantitative data can be subjected to PCA analysis to visualize the clustering potential of the chosen markers in honey [138]. 


Multiple unsupervised methods are used in some research for exploratory analysis. One example is the research presented in [139], where one way ANOVA, CA and PCA are used on IPC-MS data to explore the relation between concentration in thirty nine elements and the geographical origin of Bracatinga honeydew [139].

#### 4.1.2. Supervised Chemometric Methods

The supervised methods need an initial set of classified or labeled data to adjust the parameters (to train) of a model which is then used as a predictor. The trained predictor can classify new samples in the existing classes. The methods used for food product and in particular for honey characterization can be grouped in: DA methods which use the observations of a number of variables for each sample for the separation of samples of the training set in groups and for the allocation of new (test) samples in these groups [140]. DA methods can be grouped after the type of relation used in:
oLinear Discriminant Analysis (LDA) which builds a discriminator function as a linear combination of the independent variables. It is a common technique used to build predictors for the botanical and geographical origins of honey based on their composition. One recent example is given in [139], that used LDA to develop a predictor for the geographical origin of Bracatinga honeydew honey based on IPC-MS data.


Measurement interferences can make the separation of spectral data harder. In order to facilitate the classification of spectral data, the results of measurements are preprocessed with different mathematical techniques. The techniques mostly employed are Multiplicative Scattering Correction (MSC), normalization, Standard Normal Variate (SNV) transformation, De-Trending (DT) baseline correction, first- and second-order derivatives [141]. Light scattering present in the spectra of food is greatly reduced by MSC and SNV. By applying the first and second derivatives, the baseline and random noise are reduced. The same treatment can be used to preserve the information for quantitative determinations. Individual spectra analysis is enhanced by the use of DT. The baseline shift and curvilinearity variations can be also handled by using DT [142].

Fourier Transform Infrared (FTIR) spectral data are used for the prediction of mono-floral honey type and of honey’s physico-chemical properties [143]; the preprocessing techniques investigated are MSC, MSC + first derivative, MSC + second derivative, SNV, SNV + DT, SNV + first derivative, SNV + second derivative, first derivative and second derivative. They are compared with no treatment of the spectra. PCA is used for the study of the samples grouping according to their botanical origin. The study shows a distinct grouping for mint, rape, acacia, tilia and sunflower honeys, and mixing for thyme and raspberry honeys. An LDA-based classifier is then developed to reliably identify the link of the sample to one of the seven honey groups. The best results are obtained with the MSC pretreatment. For each of the physico-chemical parameters of interest for the honey quality, i.e., M, pH, electrical conductivity (EC), free acidity, HMF, fructose, glucose and sucrose contents, an individual PLS-R-based predictor is developed. Each predictor uses data from a specific spectral band. SNV combined with the first derivative is the best pretreatment method for predicting pH, electrical conductivity, free acidity, 5-HMF, fructose, glucose and sucrose. The first derivative is the best data pretreatment for the prediction of moisture content [143]: oStepwise Linear Discriminant Analysis (SLDA) uses a stepwise inclusion of the independent variables in the model [144].

PCA, MANOVA and SLDA are used to build a correlation between the volatile compounds’ composition measured by headspace solid-phase microextraction coupled to gas chromatography/mass spectrometry (HS-SPME/GC-MS) and the geographical provenance of Greek Quercus ilex honey [145]. MANOVA is used to identify the significant VOCs, PCA to construct the groups and SLDA to construct a predictor for the origin of a new sample [145]. 

Partial Least Square (PLS) methods are regression-type methods. In opposition to the Ordinary Least Squares (OLS), where all independent variables are used, in PLS a smaller number of uncorrelated components are generated from the independent variables in a similar fashion to PCA [146]. Some examples of using these components for regression in honey analysis are:
oSimple Partial Least Square Regression (PLSR) [147] is used on the HPLC data decomposed by ATLD to correlate the content in the phenolic compounds used as markers for the honey’s antioxidant activity [28]; oPartial Least Square—Discriminant Analysis (PLS-DA) is a combination between PLS and DA, used when categorical results are needed [148]. The influence of different preprocessing steps (autoscale, variance (std) scaling, min–max scaling, class centroid centering and scaling, smoothing, SNV and Pareto) on the accuracy of a PLS DA predictor for the geographical and botanical origin of honey, is analyzed by [149]. The predictor uses ^1^H NMR spectra data. A first pretreatment step is the reduction in the data by replacing each six consecutive chemical shifts with their mean. For the geographical origin, identification of the highest accuracy is obtained through autoscale, variance (std) scaling and class centroid centering and scaling. For the botanical origin, the highest accuracy is obtained through the variance (std) scaling data pre-treatment [149]; oUnfolded PLS-DA UPLS-DA combines unfolded PLS [150] which decompose the sample spectra to extract the relevant information with DA; oMultilinear PLS-DA MPLS-DA combines multilinear PLS [151,152] which can use multidimensional data as input with DA. 


The suitability of different DA techniques for building adulterant predictors based on fluorescence spectrometry data are compared with PARAFAC [134]. PLS-DA, UPLS-DA and NPLS-DA are used to build classification models. The UPLS-DA model performs the best and the PLS-DA performs the worst [134]: oLinear discriminant analysis based on partial least-squares (PLS-LDA) in which LDA is performed using PLS as the reduction step [153]; oOrthogonal projections to latent structures discriminant analysis (OPLS-DA) combines Orthogonal projections to latent structures (OPLS), which separates the independent variables into predictive and uncorrelated variables, with DA for a categorical response [154];
^1^H NMR spectra of Chinese honey samples are used to identify adulterated honey. A PCA LDA discriminator and an OPLS-DA one were built, trained, validated and tested. The OPLS-DA has a slightly better accuracy. The OPLS-DA also helped to identify a set of substances with significantly different concentrations in altered and unaltered honey that can be used as a marker for adulteration [155];PCA and OPLS-DA on proteomics data obtained with sequential window acquisition of all theoretical fragment ion mass spectra (SWATH-MS) to develop a predictor for honey adulteration, the producing region (Tainan, Changhua, and Taichung), country (Taiwan and Thailand) and botanical sources (longan and litchi) [138]:
oOrthogonalized partial least squares coupled with linear discriminant analysis (SO-PLS-LDA) is a multi-block discriminant classifier that results from the combination of LDA with the sequential and orthogonalized-partial least squares method (SO-PLS) which is a multi-block regression method [156,157]. A method for detection of honey alteration after heat treatment (4 h at 80 °C) is presented in [158]. The data are obtained through differential pulse voltammetry using three types of Natural Deep Eutectic Solvents (NADES) buffers and a normal buffer with the multiple wells screen-printed carbon electrodes. The data for each type of buffer were first used individually for developing a PLS DA classification models. With the fused data from the four sensors, a multiblock classifier based on SO-PLS-LDA with very good accuracy is developed; 
k-Nearest Neighbors method (kNN) classifies the sample based on the classes of the k-nearest neighbors [159];Soft Independent Modeling by Class Analogy Method (SIMCA) uses PCA on the samples of the training set for the construction of the classification models [160].

The correlation between honey sample fluorescence spectra and their botanical (acacia, linden, honeydew, colza, sunflower, chestnut, lavender) and geographical (Romania and France) origin [161] is investigated. The PARAFAC was used to decompose and visualize the fluorescence excitation emission matrices to identify the features that allow classification. On the basis of the obtained PCs, a SIMCA model was developed that predicts the geographical and botanical origin [161]. 

A comparison between two analytical methods, MIR spectroscopy and Matrix-Assisted Laser Desorption Ionization (MALDI)-Time of Flight (ToF)-based MS (MALDI-ToF-MS) and between multiple chemometric methods PCA-LDA, PCA-kNN and SIMCA for the identification of the botanical origin of the honey is presented [162]. The datasets from each of the analytical methods were used for the development of predictors with each of the enumerated chemometric methods. The SIMCA method has proven better able to identify both monofloral and multifloral honeys due to its “soft” classifier which can identify a sample as an outlier or belonging to multiple groups as opposed to the hard classifier of the other methods which classify a sample strictly in a single existing group. The MIR method due to its higher reproducibility, lower demand for manual labor and laboratory infrastructure is more suitable for implementing an automatic authentication method based on a cloud-located database that can be used worldwide [162]. 

Support Vector Machine methods (SVM) use the training set to construct the hyperplane that separates the classes with the largest margin [163]. Support vector machine can be used for regression (SVR) [163] or for classification (SVC) [164]. The Least Squared-Support Vector Machines (LS-SVM) [165] are improved variants.

The performances of four methods for building predictors that use Attenuated Total Reflectance-Fourier Transform Infrared (ATR–FTIR) spectral data are compared to identify adulteration with rice syrup in honey from three different botanical origins (acacia, linden and jujube) [166]. The four used methods are PLS-DA, Der-PLS-DA, LS-SVM and Der-LS-SVM. The methods noted with Der implement a derivative-based pre-processing for removing spectra baseline offsets before modeling. 

Hyperspectral imaging combines the spectral information from spectroscopy with spatial information from digital imaging [167]. The data from VIS-NIR hyperspectral imaging are used to develop a machine learning-based classifier for the botanical origin of the honey. The developed classifiers combines two classification methods, SVC and kNN, and attains an accuracy of 91% for close set and 80% for open set cases [168].

A linear classifier based on PLS-DA and a nonlinear classifier based on SVM for the species of honey-producing insects are presented in [169]. The data from the GCMS of honey samples were first investigated with PCA to identify the physico-chemical parame-ters and volatile compounds relevant for species’ determination. The contribution of the parameters to the clustering of the samples were identified using HCA. Both models showed good performances [169].

A new gold nanoparticle sensor array that changes the color in contact with the volatile components in honey was developed [170]. The resulting images from the analysis of honey samples from three different botanical species (acacia, canola, honeydew) are used as training set for three supervised methods LDA, PLS-DA and SVM. The best accuracy in identifying a new sample was obtained with SVM [170];

Artificial Neural Networks (ANN) are universal approximators that mimic the functioning of biological neurons [171]. Convolutional neural networks (CNN) are ANN in which the connectivity is inspired by the animal visual cortex.

The approach presented in [172] uses data from Raman spectra of pure glucose maltose, fructose and sucrose solution and from commercial honeys to train ANN to identify the honey concentrations in those sugars. Multiple network structures were mentioned. This indicates the possibility of developing a non-invasive, rapid, automatable and cost-effective method for honey-sugar composition analysis [172]. 

Physico-chemical (moisture, fructose, glucose and sucrose content) and rheological (loss modulus, elastic modulus, complex viscosity, shear storage compliance and shear loss compliance) parameters of the honey samples are used for the identification of their botanical origin. Two predictors were developed, one based on LDA and the other based on ANN, both with good accuracy. A rheological properties’ predictor based on the ANN that uses the physico-chemical parameters of honey was also developed [173]. 

A deep convolutional neural-network was used to classify honey based on the results from a multi-electrode differential pulse voltammetry. The honeys were grouped into 12 classes by PCA and then labeling for the classification was completed on the basis of the type and time of harvesting of the honeys [174].

In [175], the detection of exogenous sugars in honey based on the analysis of the Raman spectra of honey samples was investigated. PLS-DA, PCA-LDA and kNN were used to construct predictors for the quantity and type of multiple adulterants. The study was then extended with an exploratory analysis using PCA and t-distributed stochastic neighbor embedding (tSNE) [176] to correlate the results from Raman spectroscopy with the botanical origin and the possible adulteration. tSNE proved better at separating the adulterations quantitatively. Based on the exploratory analysis, predictors for both the botanical origin and the quantity of adulterants were developed on the basis of CNN, SVR and PLSR. The CNN predictor proved the best in handling the nonlinearities and multimodali-ties of the spectra followed by SVR, and the PLSR was the least accurate. Table 2 presents the chemometrics used in the honey quality analysis.

### 4.2. Modification of the Quality Parameters Used for Quality Evaluation of Honey during Processing as Analyzed by Chemometrics

The quality of honey is changed during processing and/or storage. Important physico-chemical parameters have been proposed for honey evaluation and its conformity within stipulated legislated limits [77,96], such as: water content, main sugar content (glucose + fructose), disaccharide sucrose, free acidity, diastase activity, electric conductivity, ash content, 5-hydroxymethylfurfural and water-insoluble content. Furthermore, additional attributes, associated with the basic ones above, are evaluated such as total amino acids, color parameters L*, b*, a*, amino acids composition, proline content, minerals, composition in di- and oligosaccharides, phenolic compounds, phenolic content, volatile compounds, viscosity and rheological properties, crystallization, total protein, nitrogen content and sensorial attributes. Major modifications of the physico-chemical parameters and related features by the applied changes regarding the quality of honey changes during processing/storage are discussed below in comparison with those for untreated honeys. 

#### 4.2.1. Free Acidity 

Free acidity (FA) originates from organic acids, being in equilibrium with their internal esters lactones and inorganic ions such as phosphates, sulfates, nitrates and chlorides, capable of generating their conjugated acids [36], while its determination constitutes one of the basic parameters (≤50 meq/kg of honey) stipulating limits for assuring the protection of the product’s quality from microbial activity [77]. FA is attributed to: (a)Organic acids. Acidity is mainly derived from the presence of organic acids, up to 0.5% in honeys, contributing to honey flavor, stability against microorganisms, enhancement of chemical reactions and antibacterial and antioxidant activities [178]. The principal organic acid in honey is gluconic acid derived from the activity of the glucose-oxidase enzyme on the glucose substrate, that is in equilibrium with δ-gluconolactone [179,180,181,182]. The gluconic acid level, for a specific honey species, is mostly dependent on the time elapsed between the collection of nectar and formation of the final honey by bees for obtaining the final density in the honeycomb cells, while glucose–oxidase activity becomes insignificant when the honey is thickened [183]. Moreover, other organic acids are found in honey such as formic, aspartic, acetic, butyric, citric, fumaric, galacturonic, gluconic, glutamic, butyric, glutaric, 2-hydroxybutyric, glyoxylic, α -hydroxyglutaric, lactic, isocitric, α-ketoglutaric, malic, 2-oxopentanoic, malonic, methylmalonic, propionic, pyruvic, quinic, shikimic, succinic, tartaric, oxalic acid and others [184]; their ratio and abundance are influenced by the honey species enabling discrimination of the honeys [179,185], while some organic acids have exhibited a high discriminant power for the separation of conventional from organic honeys [186];(b)Lactones. Lactones found in honey are mostly in the form of gluconolactones, constituting part of the organic acids in the intra-esterified form; they contribute a reserved acidity measured when the honey solution becomes alkaline [36]; lactonic acidity is added to FA to yield the total acidity of honey [77]. The pH of honey and its acidity are not parameters directly related to each other because many other components found in honey exert a buffering capacity, therefore, compensating for a part of honey’s true acidity [187,188]. Similarly to pH, free and lactonic acidity in the different honeys are dependent on their botanical origin, also influenced by the harvesting season [178,183,187,189,190,191].

During storage time, the amount of organic acids significantly increases [192]. Several factors contribute to honey FA:(a)Effect of maturation. During honey maturation, the FA or total acidity is increased while pH is significantly decreased [178]. In a pioneering study covering the introduction of national legislative limits for Talh honey, the free acidity (FA) of Talh honey was determined from Talh tree leaves and flowers (30 ± 0.99; 34 ± 0.92 meq/kg) to bee crop (honey stomach) and unripe honey (43 ± 1.80; 72 ± 1.56 meq/kg) and finally to ripe honey (77 ± 1.28 meq/kg), [193], while the highest pH value was recorded in the leaves and kept decreasing as honey production proceeded, obtaining its lowest value in ripe honey (4.91 ± 0.06);(b)Effect of storage. Reports have shown a significant effect of storage on honey FA, pH, (*p* < 0.05), with FA increasing and pH decreasing with storage time [181,194,195]. In one kinetic study, exclusively dedicated to the variability of all the three parameters versus 30 months storage for honey stored at room temperature (15–25 °C), lactonic acidity found to increase by storage time (*p* < 0.05), even at a higher degree than FA increased or pH decreased [196], while in some cases lactonic acidity was slightly decreased, and total acidity was increased [181]. Formation of levulinic and formic acids also is derived from 5-HMF transformation, and keep increasing by storage [197]. Evaluation of the variability of FA, pH, lactonic acidity, and total acidity has resulted in estimation of 20 months of storage to be the “best before” period “once opened” [196].Investigation of the effect of short storage at 35–40 °C for 3 and 6 months with or without the addition of metabisulfites (12 pp) on water content (WC), pH, FA, lactone acidity and total acidity of two honeys, cashew and marmeleiro [198], showed that significant differences were observed for pH, FA, lactone acidity and total acidity compared to the respective parameters for the fresh samples. A reverse correlation between FA and lactone acidity was recorded and attributed to the glucose–oxidase activity that converts glucose to gluconolactone, which is consequently hydrolyzed to gluconic acid. In this study, FA is reduced but lactone acidity is increased with the storage time. The presence of bisulfite acted upon the esterification of gluconic acid to increase the lactone concentration [198]; (c)Effect of dehumidification. Dehumidification of honey in other studies has shown no differences in pH and FA between raw and dehumidified honeys when a group of samples from the stingless bees *H. itama*, *G. thoratica* and *T. apicalis* honeybee species were used. However, samples of *H. itamas* honey had a lower FA and higher pH and ash content values than *G. thoratica* honey samples [199], similarly to honeys of the other bee tribe [200]; (d)Effect of temperature/storage. Storage under different thermal conditions for times up to eight months induced a great increase in the free acidity of Talh honey, a rare type of honey because of its high FA. Talh honey naturally exceeds the permitted level for the FA values (>50 meq/kg) which is attributable to the plant origin. Storage temperature was found to be a factor with the highest significant influencing power on the FA (*p* < 0.05). Although all the values of FA in this study were beyond the standard limit, the results indicated that the stability of the FA of Talh honey was maintained stable at low temperatures (0–25 °C) for up to 6 months without significant effects [194]. In this study, statistical analysis showed the FA to exert a positive correlation with storage period (0.401), storage temperature (0.631), 5-HMF (0.852), color (0.541), moisture (0.440) and EC (0.155). On the other hand, FA was negatively correlated (*p* < 0.05) with glucose (−0.892), pH (−0.851), fructose (−0.821), sucrose (−0.422) and diastase activity (DN) (−0.309). Thus, low pH, DN and sugars are associated with higher FA. The strong positive correlation of FA with the 5-HMF is related to the strong effect of pH on the formation of furfurals generated more by the Amadori Rearrangement Products pathway than the routes of reductones and fission products dominant at pH > 7 [181,201].

#### 4.2.2. Ash Content and Electric Conductivity (EC)

Ash content is indicative of the amount of minerals contained in honey [191]. There is a great dependency of ash content on the type of soil, climatic conditions and environmental pollution. The dissolved salts in the soil are pumped to the flowers used for nectar collected by the bees [31,32], enabling ash content to be a good indicator of the geographic origin of honey [49,202]. Mineral content is strongly correlated with the color and EC, affecting the color and flavor of the honey. Honeydew contains a higher quantity of minerals, therefore, it is commonly used in quality control to distinguish honeydew from floral honey [202,203], while citrus has a lower EC [204]. EC is influenced by the presence of salts, organic acids, minerals, amino acids, proteins, storage time and different sources [187]. The legislative limits require EC ≤ 0.8 mS/cm for all honeys, and ash content ≤0.6 in general and ≤0.1 for honeydew honey and its blends with blossom honey [77,96].

Storage and/or thermal treatment of honey results in an increasing effect of EC on honey [204]. Correlation analysis showed a strong positive correlation (*p* < 0.05) between EC and storage period, HMF and FA, whereas, a negative correlation (*p* < 0.05) was recorded between EC and sucrose, glucose and fructose [192,194,205,206].

#### 4.2.3. Sugars 

Carbohydrates are the main constituents of honey [207]. They are produced from nectar sucrose by honey-bees, which is transformed through the catalytic action of several enzymes, mainly α-, and β-glucosidase α- and β-amylase, and β-fructosidase, diastase and invertase, resulting in a composite mixture of monosaccharides, disaccharides and oligosaccharides [183,208]. Glucose and fructose monosaccharides constitute the major honey saccharides, ranging from 65% to 80% of the total soluble solids, followed by disaccharides and trisaccharides, while more than 26 sugars have been identified and quantified in honey [207]. The composition of the carbohydrate fraction of honey is strongly dependent on the plant species from which the nectar is collected by the bees, the bee species, maturation, with geographical and seasonal effect being negligible [183,209]. It has also been reported that bee species have a strong effect on the differentiation of honeys as has been revealed by NMR studies based on the different conformers of glucose and fructose contained in honey [210]. 

Fructose and glucose are the predominant sugars in honey, with fructose found in higher amounts except rape *(Brassisa napus)*, blue curls *(Trichoderma lanceolatum)* and dandelion *(Taraxacum officinale*) honeys, where glucose is in a higher quantity [98,207]. They are derived from the transformation of nectar’s sucrose via the enzymatic action of invertase contained in the salivary glands of bees. However, invertase also owns transglucosylation activity, catalyzing the α-glucosylation of monosaccharides, disaccharides or trisaccharides in honey resulting in the formation of di- and tri-saccharides [203]. More than 30 saccharides have been identified, such as: (a) di-saccharides: sucrose or saccharose (predominant), maltose, turanose, cellobiose, kojibiose, maltulose, trehalose, nigerose, isomaltose, trehalulose, gentiobiose, laminaribiose, palatinose, gentiobiose, (b) tri-saccharides: erlose, theanderose, panose, maltotriose, 1-kestose, isomaltotriose, melezitose, isopanose, gentose, 3-α-isomaltosylglucos, planteose, (c) oligosaccharides: raffinose, isomaltotetraose, isomaltopentaose.

Enzymatic α-glycosylation of monosaccharides to di- and tri- saccharides starts once the mixture of nectar enriched with bees’ saliva is placed in combs by the bees. A recent study reports that the acacia honey has ripened when the concentration of turanose is above 1.2 g/100 g honey, which occurs just after the combs are capped by the bees, timing before the 10th day of honey ripening in the honey combs [211]. A strong positive correlation has been recorded between the days of honey maturation in combs and the turanose content (*p* < 005) but a strong negative correlation with the water content as expected to occur during maturation in the beehive [211].

The amount of sucrose is dependent on its botanical source, honey maturity, elevated nectar abundance or artificial bee-feeding [59,183,212]. The steps in honey maturation involve an appreciable decrease in sucrose because of the continuous action of invertase added by the bee. Therefore, the maximum limit of sucrose (<5 g/100 g) is an indicator of freshness or possibly adulteration [134]. The evolution of main honey monosaccharides, glucose, fructose and disaccharides has been investigated by the fractionation of stable carbon isotope (δ^13^C) towards the examination steps for: (i) flowers, (ii) stamens, (iii) nectar and the (iii) ripened rape honey. It has been reported that δ^13^C keeps increasing with the same order with these steps reaching a maximum for the ripened honey, which is significantly different than the δ^13^C of the previous steps, a fact attributed to the addition of enzymes for sucrose inversion and the water evaporation caused by the bees’ fanning with their wings [213].

During prolonged storage of honey, the amount of fructose, glucose and sucrose decrease. A remarkable decrease of 9% of monosaccharides per year was recorded during prolonged storage [107].

Reported studies for Tahl honey, which by its nature has FA values exceeding international limits, recorded a high decrease in saccharide concentrations especially for samples stored at 35 and 45 °C for different time intervals up to eight months at 0, 25, 35 and 45 °C [194]. Under all the storage temperatures, the sugars (fructose, glucose and sucrose) significantly decreased during the storage period with reducing sugars’ amounts to reach levels below legislation limits (Fructose + Glucose >60 g/100 g) [96]. This declining trend for sugars is more pronounced at higher temperatures, 35 and 45 °C. Based on the results, temperatures below 25 °C are suggested for maintaining Talh honey. Statistical analysis showed higher Pearson’s coefficients among parameters: positive correlations between sugars and pH and DN, but between sugars and storage period, temperature, color, EC, HMF, FA and moisture negative correlations were recorded.

Very important losses in the monosaccharides of citrus honeys were recorded during storage at different temperatures for 12 months reaching at 13.5, 25 and 25.2% for 10, 20 and 40 °C, respectively, compared to those of fresh honeys [195]. Sucrose is significantly decreased. However, marked increased changes were recorded for other disaccharides, such as nigerose, turanose, maltulose, isomaltose and kojibiose, whereas trisaccharides did not show any trend. In the same line, maltose with an initial amount of 2.5 mg/g changed to 23.2 mg/g after 1 year stored at 40 °C, predominating all the saccharides. Both the presence of invertase, which also acts as transglycosidase on glucose substrates, and the low pH values arising from the elongated storage enhance the conversion of monosaccharides to disaccharides and higher sugars, and count for sucrose decrease and maltose increase in the thermal-treated honeys [194,198,214,215,216]. High levels of maltose could be used as an indicator of the prolonged storage of honey [195].

Use of thermo-sonication for the honey dehydration processing led to a higher increase in 5-HMF compared to conventional thermal processing. Higher dehydration rates recorded for the thermos-sonication method than bath process count for higher formation rates of 5-HMF as an intermediate product of acid-catalyzed dehydration reaction of hexose and/or by Maillard reaction than conventional thermal treatment [217]. Although the rate of 5-HMF for conventionally heated honeys was the largest for acacia honey, during the processing by microwave the fastest formation of 5-HMF was recorded for lime honey [218]. The conventional and microwave heat processing induced the largest relative increase in the 5-HMF formation in honeydew honey while other studies have reported that during the microwave heating the most rapid increase in 5-HMF was found in lime honey; all these results indicate that the formation of the 5-HMF is dependent on the particular composition of each honey [108,218].

Chemical composition changes for sugars have been studied during thermal treatment and/or storage using FTIR spectroscopy as a tool for mining data [219]. Prominent changes have been recorded for the region 900–1500 cm^−1^ [220], with specific absorptions at 987 and 1040 cm^−1^ where maximum variances were recorded originating from carbohydrates absorption, mainly fructose, glucose, and sucrose [221]. Chemometric discriminating methods applied on FTIR data ranging from 600 to 4000 cm^−1^ discriminated between raw and thermally treated honeys. Furthermore, by using chemometrics the honeys were categorized for (a) 70 °C, for samples treated for 15 min and those for 120 min (overall accuracy 0.947%); (b) 40 °C, for samples treated for 15 min, and those for 120 min (overall accuracy 0.895%), leading to the conclusion that the model is successful for classifying the samples according to different thermal treatments based on the carbohydrate’s changes.

#### 4.2.4. HMF

Heating, dehydration, and storage processes play an important role in the formation of 5-HMF. This furanic compound originates from different chemical pathways occurring during honey processing (a) from dehydration of hexoses in acidic conditions [222], and (b) as an intermediate of the Maillard reaction [195,197,223].

The fact that the 5-HMF mostly reaches low concentrations in fresh honeys means that it is commonly employed as a quality parameter for assessing the freshness and/or overheating of honey. Legal regulation bodies have set a maximum 5-HMF content of 40 mg kg^−1^ for honeys in general, and 80 mg kg^−1^ for honeys from tropical climates, including blends [77]. Furthermore, there are safety concerns by consumers for foods of high 5-HMF content, such as DNA mutagenicity and colon carcinogenicity among others [224,225,226]. 

Honey freshness based on 5-HMF can be determined with several analytical protocols, however, an easy determination is executed using the ^1^H NMR spectroscopy, where the integration of signals occurs at 4.65 ppm for the methyl protons or at 9.55 ppm for the proton of aldehyde, enabling its quantification [227].

Storage of cashew honey for up to six months resulted in the excess formation of 5-HMF at levels exceeding the legal limits. Addition of metabisulfite in honey at the start decreased the 5-HMF concentration, a fact attributed to the formation of sulfonic acids of the dehydro-reductone Maillard intermediary interrupting the cyclization prior to the formation of 5-HMF [198]. The 5-HMF can be produced from all hexoses, but actually is selectively derived from keto-hexoses, such as fructose because of (a) enolization of fructose proceeds with a comparatively higher rate than that of glucose because in solution fructose forms less stable ring structures enabling it to spend a larger life-time in the open chain form and (b) fructose becomes involved in an equilibrium reaction by forming di-fructose-di-anhydrides so that the most reactive groups capable for cross-polymerization are intra-disabled favoring an increase in selectivity, whereas glucose forms real oligosaccharides which still have reactive reducing groups, available for cross-polymerization with HMF and reactive intermediates [228,229,230].

There are many factors influencing 5-HMF levels, such as temperature, storage conditions, water activity, divalent cations concentrated in the media, the flowers’ origin, and some chemical properties of honey, including pH, acidity, reducing sugars, and mineral content, of which the pH is the most recorded [195,218,228,231]. For honeys which have an unusual high FA, exceeding the legislative limit, a great increase in FA was followed by a very high increase in 5-HMF, which was more pronounced for honeys stored at 35 °C for twelve months, and for those stored at 45 °C from the very first month. A very strong positive correlation of Pearson’s coefficient found between 5-HMF and FA, storage time, and storage temperature [194]. Reports on thermal formation kinetics of 5-HMF fitting to a first- or zero-order reaction, or first-order kinetics of the degradation of amino acid and zero-order kinetics for the formation of product (5-HMF), resulted in activation energies calculated according to the Arrhenius model in various honeydew and floral honeys evidencing that, besides the processing treatment, the composition of honey also has a role on the 5-HMG formation [223,228,232,233,234].

The effect of thermal processing in the content of 5-HMF in monofloral pine, citrus, thyme, and eucalyptus, blends (Thyme-pine, Erica-pine) and multifloral honeys from Crete, at different temperatures and storage times showed that the 5-HMF changes for Pine honey are the most resistant towards thermal treatment especially for high temperature and prolonged times while thyme was the most vulnerable to 5-HMF increases, followed by citrus honey, a fact attributed to the higher pH value of pine honey [190].

Separation of raw honeys from those heated at different temperature levels was achieved using FTIR data and chemometrics. PCA analysis for eucalyptus honey yields a strong discriminating factor for complete separation of raw samples from those heated at 40 °C/3.5 h and 70 °C/15 min, but not for acacia or orange blossom honeys, based on the spectral features at 990 and 1050 cm^−1^, where mostly stretching (C-OH) vibrations of carbohydrates occur [221]. Application of DA showed sufficient categorization for (a) raw samples heated to 70 °C for 15 min, and heated samples to 70 °C 120 min (overall accuracy 0.947%) (b) raw heated samples to 40 °C for 15, and samples heated to 40 °C for 120 min samples (overall accuracy 0.895%), thus, enabling classification of the samples according to different thermal conditions, temperature, and duration [221]. The effect of storage has also been investigated with other spectroscopies such as ^1^H NMR spectroscopy, where mainly sugars and minor components were found to differentiate honeys [29].

During storage or thermal treatment furanic compounds are formed besides 5-HMF, such as 2-HMF and 2-(furan-2-methyl)-4-methoxyfuran-3-(2H)-one (2-Furfural). These are derived by Amadori degradation via: (a) enolization for pH < 7 with 2-HMF formation to prevail (implication of pentose, as xylose is implicated) and (b) enolization at basic pH, 5-HMF (implication of hexose, as glucose) could be produced through reductones and the fission products’ route [181,201,235]. In honeys owning mostly an acidic pH, 5-HMF prevails whereas in wood and chestnut honeys, 2-Furfural is predominant because of the higher pH values [236], however, different conclusions have been reported as well [197]. The Maillard reaction is commenced by nitrogen-containing compounds, and carbonyl compounds such as amino acids in foods resulting in Schiff base intermediates and rearrangement to Amadori or Heyns products. The α-Dicarbonyl compounds (α-DCs), which are highly reactive critical intermediates in the Maillard reaction [237,238], are affected by pH, nectar composition, and storage. Artificially matured honeys significantly exceed in α-DCs those that are naturally matured [237], proving α-DCs to be a more sensitive indicator for heat treatment than HMF [237,239].

#### 4.2.5. Components in Crystallization

Crystallization and viscosity affect sensorial properties of honeys, in consequence, the consumers’ acceptability. Heating pretreatment delays the crystallization of honeys while maintaining low viscosity values, with the most common methods of crystal prevention including the pasteurization method (treatment at high temperatures), storage at low temperature, microwave or ultra sound pre-treatment, storage at very low temperature and filtration using sieves (pores < 80 μm) [34,38]. Heating procedure at 70 °C/15 min manages to extend the non-crystallization period, but it negatively affects the quality of the honey and its shelf life [240].

The ratio of fructose to glucose (F/G) and their relationship with water content govern the rate of honey crystallization, and as a consequence, have a predominant role for controlling the rheological properties of honeys [241,242]. An effect of the botanical source on the honey crystallization and stability has also been recorded [243]. Molecular dynamics revealed that the crystallization of honey was different from that of pure glucose regarding the morphology and conformational stability, with glucose/fructose at 2.5:1 to result in crystallization of the same stability as the crystals found in honey described of the same glucose/fructose ratio [244].

The parameter Fructose/Water (F/G) can predict crystallization because glucose is less soluble than fructose. When F/G < 1.4, fast crystallization of honey occurs but when ratio is > 1.54 honey exhibits no tendency to crystallize [245,246]. Strong relationships among sugars (fructose, glucose, sucrose, melezitose, and maltose), palynological characteristics, sugar ratios (F + G, F/G, G/W), and moisture content were revealed when statistics applied towards the establishment of a predictive base for the crystallization tendency of monofloral honeys [98]. PCA discriminated rape and sunflower honey that exhibited the highest F + G mean values (>75%) which are significantly different compared to honeydew, eucalyptus, bramble, heather, chestnut honeys (*p* < 0.05). Honeydew exhibited the lowest F + G concentration (< 60%) than the others (*p* < 0.05). This parameter is a marker for distinguishing honeydew honey from blossom [182]. Rape and sunflower, being blossom honeys, are known from the literature to easily crystallize during storage, therefore, they are expected to have the highest F/G ratio in the PCA components plot [98]. Water content is another parameter that affects crystallization. The ratio G/W < 1.7 is indicative of slow or no crystallization but ratio > 2 for complete or fast crystallization [107]. The higher the glucose content and the lower the water content, the higher the crystallization rate proceeds. Many sugars crystallize if their concentration is above the saturated level [205]. Glucose in high concentration leads honey to crystallization because it is less soluble and crystallizes faster than fructose.

The effect of the phase of honey during thermal treatment conditions on the moisture, 5-HMF content, lightness, and yellowness has been recorded, where moisture loss of crystallized and Bi-phase honeys are greatly affected by the thermal treatment at a low temperature of 39 °C/30 min and at a high temperature of 55 °C/24 h, while liquid samples were not affected [247]. Chemometrics were applied on FTIR spectroscopic data, which express structural modifications induced by the heating process originating from the dissolution of glucose crystals, acquired at 1470 (O-H stretching vibrations), 1935 (H_2_O-OH bonds of water molecules), 2100 (carbohydrates), and 1690 cm^−1^ for fructose, for highlighting the ratio on F/G, showing (a) a high predictive ability (0.78 MMC) to discriminate honeys treated at room temperature from those at high temperature. Conventional and chemometric approaches showed that changes in physico-chemical parameters and NIR spectroscopic characteristics were larger for crystallized and Bi-phase honeys than for those in liquid phase, therefore, they are phase related [247].

Prediction for the type of liquefaction treatment can be achieved for honeys liquefied at 40 °C or 72 °C first and afterwards stored for 12 months at different temperatures, based on changes in the physico-chemical parameters associated with the crystallization occurrence in honeys versus time [240]. A PCA statistical model showed separation for each group of samples with each one subjected to different liquefaction treatment, while HCA categorized samples into three groups, one for heated honeys at low temperatures (0–18 °C), a second for unheated (fresh + heated at a low temperature), and a third group containing the samples heated at a high temperature level [240]. Samples stored at −18 °C showed no difference in the physico-chemical parameters from the fresh ones; it may affect honey viscosity [220].

Dehydration of honeys at 40 °C goes through several steps: at the two first steps, water loss occurs, the third step is characterized by volatiles’ loss, whereas, at the last dehydration step both volatiles and water are removed from samples [248]. Kinetics of honey heating at 40 °C for time intervals varying up to 90 h were performed with Synchronous 2D correlation spectroscopy for monitoring changes at 995 nm for rape and 990 nm for chaste honeys versus increasing dehydration times. NIR spectra showed differentiation at 900–990 nm for absorption assigned to the second overtone stretching vibrations of NH of aromatic amines and OH of CHOH groups originated from sugars, enzymes, fragrant aroma, water, and other components of honey [249].

The presence of oligosaccharides can modify the honey’s tendency for crystallization [98] Honeydew honeys containing a higher content of melezitose do not easily crystallize [203,207,250]. In this line, crystallization prevention has been reported based on the addition of trehalose to the honey samples, a fact attributed to trehalose’s high hydrophilic character that prevents the formation of internal hydrogen bonds but enhances the tendency to form hydrogen bonds with surrounding macromolecules resulting in preventing glucose crystallization and the expulsion of water molecules from its supersaturated solution [251]. Although the addition of trehalose leaves the G/W ratio the same, trehalose interferes with the dynamics of water molecules by hindering glucose from crystallizing in the supersaturated honey solution [252].

The crystallization process is directly associated with the chemical composition of honey [98]. The ease with which glucose monohydrate forms crystals in honey is due to its ability to adopt miscellaneous geometric forms leading to a change not only in the consistency but also in the water binding during the crystallization process [85,246]. A recent approach, developed for detecting honey liquefaction based on (i) the water impact on pH and the monosaccharides and (ii) disaccharides’ contents of honey, predicts if honey has previously undergone liquefaction above 30 °C based on the modification of EC behavior and pH changes, both induced by the irreversible changes of the molecular structure [107,205].

#### 4.2.6. Amino Acids/Proteins

Amino acids are contained in honeys accounting for 1% (*w*/*w*) [13]. They are originated mostly from pollen, and in a lesser degree from animal and vegetable sources, among them fluids and the nectar secretions of salivary glands and pharynx of bees [253]. Several amino acids are contained in honeys such as arginine, aspartic, serine, glutamic, asparagine, glutamine, threonine, proline, phenylalanine, histidine, β-alanine, glycine, α-alanine, γ-aminobutyric, tyrosine, α-aminobutyric, tryptophan, methionine, valine, isoleucine, leucine, ornithine, and lysine [253]. The composition of amino acids in honey is dependent on the pollen species from where the nectar originates, therefore, it can serve as an index for botanical differentiation [56,254]. Proline, originated from the salivary secretion of bees acting on honey collection and maturation, has been found in higher amounts than the other amino acids in honeys, as the major contributor with 50–85% of the total amino acids [208,255,256], while in some studies, phenylalanine has been reported to be the predominant one [253,257]. In general, honeydew contains higher amounts of amino acids than floral honeys [203], and can be differentiated from raw floral honey by amino acids composition in all amino acids except for phenylalanine, tyrosine, and proline [255]. Chemometrics on the amino acid profile of honeydew *Mimosa scabrella Bentham* species resulted in differentiation based on geographical origin [57].

Proteins such as diastases (α-, β- amylases) for hydrolysis of a-D-(1,4) glycosidic bonds, invertase for sucrose inversion to equimolar amount of glucose and fructose, and glucose-oxidase for oxidation of glucose to δ-gluconolactone to final hydrolysis to gluconic acid, are the most common enzymes in honey [180,198,258].

The concentration of amino acids is reduced during the thermal treatment of honey or prolonged storage [203,233]. Several compounds are formed from the reaction of the carbonyl group of a reducing sugar with the free amino group originated from amino acid or protein (Maillard reaction) resulting in the formation of dark compounds followed the initial step of Amadori compounds, darkening the honey’s color. Amino acids found to easily trigger the Maillard reaction are lysine, proline, γ-aminobutyric acid, and arginine [192].

The effect of storage at room temperature on the composition of amino acids of honeydew, floral, and blends of honey for different time intervals up to 24 months showed the total amino acid content to decrease with storage, with losses ranging from 45 to 52% for glutamic acid, arginine, and glutamine in honeydew honey [258], while in Brazilian honeydew, the “Bracatinga” glutamic acid, the most sensitive to prolonged storage, decreased by 84% compared with fresh honeys [192]. In general, honeydew honeys have a higher content of individual amino acids [203,259] than floral honeys except for histidine, tyrosine, phenylalanine, leucin, and proline. Although the amount of each amino acid decreases over a storage period, their concentration and ratio in 24 months-stored honeys can still discriminate the honey type (88.7% correct assignment) [255]. The fact that the order for honeydew or floral amino acid concentrations, found in raw honeys as glutamine < asparagine < aspartic acid < glutamic acid << proline, is different than the one in stored honeys is evidence for a selective decrease in amino acids because of the different sensitivity of each of them to trigger the Maillard reaction [192]. Although it is expected that the proportion of each amino acid in processed honeys varies a lot since its reactivity is different, the loss of amino acids occurs in a way associated with the type of honey (honeydew, floral, blends) [255]. Results reported for jujube and chaste honeys showed a strong correlation of the decrease in the concentration of each amino acid for different heating treatments with the increase in 5-HMF formed during storage time. Low pH favors the Maillard reaction: it was found that chaste honey, with lower pH and higher total amino acid content, undergoes reactions of high rates for both the disappearance of the dominant amino acids and the formation of 5-HMF which were found to be several times faster than the respective reactions recorded for the jujube honey [257].

Heating or long-term storage of honey has been associated with the formation of furosine (2-furoylmethyl-lysine) generated by acid catalysis of Amadori compound fructosyl-lysine, which has been proposed as a useful indicator of the extent of the degradation of foods because of their exposure to processing or extended storage [260,261]. In addition, 2-furoylmethyl adducts of arginine, gamma-Aminobutyric acid (GABA), and proline have been detected. Furosine content is higher in honeydew honey than in other commercial/fresh honey as derived from the higher content of honeydew in amino acids, and its increase is associated with the increase in 5-HMF [260]. Samples may be considered properly heat-treated, when the furosine concentration is low, while the absence of 2-furoylmethyl-amino acids could be an indicator for honey freshness or a short period of honey storage under proper conditions [260].

Derivatives of 5-HMF, N-(1-deoxy—1-fructosyl) phenylalanine, (Fru-Phe), formed through the Amadori pathway towards thermal processing and storage of honeys, can serve as indicators of thermal process and storage. In this line, in Artificial Mature Acacia Honeys Fru-Phe is rapidly increasing, reaching 2.4 times the initial concentration while for Natural Mature Acacia Honey the increase was only 1.2 times, with respect to the initial concentrations at the start of the 24 months’ storage period [262]. Chemometrics were performed to screen for differential compounds that distinguish the two groups resulted in several compounds, however, only Fru-Phe was the one chosen because it fulfills the criteria: (a) to be present in all Artificial Mature Acacia Honeys and be in low concentrations or absent in Natural Mature Acacia Honey samples, and (b) to be stable and easily determined [237].

Reports have pointed out that D-amino acids are formed during heating of aqueous solutions of L-amino acids in the time course of the non-enzymic browning through formation of stable Amadori compounds, upon which racemization occurs [263], with the racemization to be dependent, particularly, on steric hindrance and electronic properties of the amino acid side chains [236].

#### 4.2.7. Diastase Activity

Heating processes for crystallization prevention and improvement of rheological properties of honey and elongated storage time result in denaturation of the contained enzymes as directly calculated by enzyme activities [107,195,264] or by the loss of ability to inhibit biofilm formation of certain bacteria species during thermal liquefaction of crystallized honeys [107].

Diastase is the most thermal tolerant among honey enzymes, therefore, its presence is the criterion of freshness. According to the legislation limits, diastase activity (DN) in honey in general must be ≥ 8 (Schade scale), except for honeys with low natural enzyme content (e.g., citrus), where diastase ≥3 which is accompanied by a limit for 5-HMF not exceeding 15 mg/kg in order to ensure that the low DN value does not originate from heating or elongated storage of honey [77]. Diastase hydrolyzes starch and dextrin to oligosaccharides, and exhibits activity significantly positively correlated with invertase activity [265]. It originates from pollen digestion by bees [208] and from nectar (blossom and honeydew), and therefore is dependent on the botanical origin [264,266]. It is also dependent on the age of the bees, the nectar harvesting season, the physiological period of the colony, the large quantity of nectar flow, its sugar content since a high flow of concentrated nectar results in a decrease in enzyme concentration and pollen consumption [228], and the bee tribe [267] while low diastase activity occurs in artificial honeybees feeding [59]. Consequently, DN alone is not an indicative parameter for honey overheating treatment [77].

High heating temperatures decrease diastase content [194,198,268], with up to 40% reported for the storage of Greek honeys for one year [59]. Chemometrics showed a negative correlation (*p* < 0.05) for diastase activity with storage time, storage temperature, FA, and HMF [194], while the diastase content decreased concomitantly with heating at higher temperatures [264].

Studies on the effect of heating temperature on the denaturation of diastase have shown a different behavior of diastase activation towards isothermal or transient patterns of heating; specifically, after isothermal treatment, diastase activity remained constant, even increased, for heating periods 600 and 1200 s for each specific temperature [190]. This peculiar behavior has been explained on the basis of the Eyring theory regarding the enzymatic recovery after the enzyme returns to the native-like state for not succeeding in overcoming the energy barrier opposed by the isothermal pattern heating [228,269]. Pine honeydew exhibits higher change-resistance heating followed by citrus, multifloral, thyme, and cotton honeys. Storage for an elongated time decreases diastase activity in honeys. Kinetics regarding storage time showed that diastase activity of pre-heated honeys from different origins were significantly different (*p* < 0.05) at both the initial and ending storage time, while the correlation of honey origin with heating exposure period was significant (*p* < 0.05) [270].

#### 4.2.8. Water Content 

The water content of honey is dependent on different factors such as the botanical and geographical origin of nectar [182,202,203,249,271], harvesting season [178], intensity of nectar flux [36], soil and climatic conditions, bee tribe [199], degree of honey maturation in the beehive, and practices followed in harvest and extraction [183]. Some properties of honey such as viscosity, density, crystallization, color, and flavor are influenced by the water content [246].

However, it is not the water (or moisture M) content but the water activity that is responsible for the quality and process attributes of honey, since water activity represents the fraction of water not tightly bound to solids (mostly sugars) contained in honey, which becomes available for yeasts’ and bacterial growth [203,271]. There is a linear relationship between water activity and water content [203,271], therefore, it is reliable to use the term water content /moisture instead. The water content of honey must be kept low, ≤20%, except for heather ≤23%, as stipulated by the legislation limits [77]. In this line, water content is deliberately reduced by dehumidification to lower than a 20% level to prevent fermentation, especially in honeys produced by stingless bees which produce honeys of a water content higher than the above limits [199]. 

During the ripening process of honey, the moisture keeps decreasing with the storage beehive cells to be totally capped with beeswax when the percentage of water in honey is appropriate [183]. 

The effect of honeycomb material on water evaporation and the maturation of honey inside the beehive has been investigated by the replacement of beeswax by a composite material consisted by 90% paraffin and 10% beeswax, and used along with the pure beeswax honeycombs within the same beehive [272]. Results showed that the water content of honeys ripening in the paraffin-based honeycombs was significantly higher than those in the beeswax honeycombs, negatively influencing the honey ripening. This was attributed to the hydrophilic character of the function groups of esters and unbound aliphatic alcohols and acids, and the presence of lipolytic enzymes incorporated into the wax by bees during comb construction enabling cells that allow for moisture transfer. ^1^H NMR spectroscopic data showed that honeys of elevated water content in paraffin combs were accompanied with a higher concentration of acetic and citric acids [272].

Water content is important for the phase stability of honey, with the ratio G/W to be proposed for the prediction of honey crystallization [98,245]. Honeydew honeys exerting a G/W ratio > 1.7 are prone to crystallization compared to blossom honeys possessing a ratio between 1.17 to 1.27 that make them vulnerable to granulate [271].

Water activity is increased during the crystallization of stored honey, because reduction in the water molecules bound to glucose occurs, from five in liquid to one in the crystalline phase, releasing water and causing the liquid phase to become less concentrated, in consequence, leading to an increase in the water activity [107,205,243,244]. On the contrary, thermal liquefaction of honey results in a decrease in water activity of honey [107]. Moisture exerted high discriminant power for the differentiation of honeys of different origin based on the storage period and botanical origin after they were first liquefied at 40 °C/60 min, and afterwards when stored at 14 °C for 60 and 180 days [243]. Significant differences (*p* < 0.001) have been reported for the water content of honeys of three different phases to follow the order: crystallized > bi-phase > liquid, at each different storage temperature [247].

Results derived from this bibliographic search shows that experimental data give a wealth of information containing hidden trends and correlations among variables that cannot been realized with other ways than chemometrics [273]. Spectroscopic methods, most commonly NMR and FTIR, and chromatography, providing the chemical fingerprint of honey are ideal to be utilized for honey chemometrics [273]. Although NMR spectroscopy has not been extensively used for honey processing and storage, the FTIR spectroscopy is currently the most common data platform for application chemometrics for quality control as well for exploring the maturation, the aging, and the degradation of honey versus processing methods and storage conditions towards quality control the preservation and/or the improvement of honey quality. In this way, it is possible that the useful information is extracted and separated from the non-useful one, also solving the problem of spectral noise. As described, honey is a complicated chemical system, with a number of parameters to influence its quality and safety, even more so since these variables are strongly correlated to each other through chemical interacting pathways and system in equilibrium states; therefore, chemometrics, as a method for multivariate data analysis, can be used for fingerprinting analysis and chemical profiling for honey affected by processing and storage.

### 4.3. Chemometrics Used in Recent Studies Related to Honey Quality Analysis during Storage and Processing

In this part of the review paper, the combination of chemometrics with analytical techniques mentioning the quality characteristics in honey during processing and storage are discussed. Specific emphasis is given to the chemometric methods used in each study and the outcome after their application. A brief presentation of very recent studies available in literature is shown in Table 3.

Segato et al. (2019) studied how heating treatments may change various physicochemical characteristics and color of honey samples in three phases (liquid, bi-phasal, and crystallized). NIR measurements were taken, and interpretation took place by applying PCA. Only the first two principal components (PCs) were important. PLS-DA and SVM have been applied on the NIR data. They worked with a training set and all the remaining samples formed the blind test. Cross-validation was conducted using a leave-one-out procedure. The three most important absorbance bands were found to be around 1420, 1905, and 2130 nm. The PCA results showed that neither mild heating nor overheating resulted in changes of the NIR data of the liquid samples; thus, they were not sensitive to temperature changes. However, overheating at 55 °C for 24 h affected the native conformation of glucose crystals in bi-phasal samples, and reduction in moisture and an increase in the HMF content, as well as a strong color change (intense browning) generated. The highest temperature strongly affected the NIR data of the crystallized and bi-phase honeys. The importance of chemometrics was underlined again, this time by SVM which showed that mild heating (39 °C for 30 min) did not affect the NIR data making this combination suitable for pre-treatment analysis if needed [240].

Oroian et al. (2017) studied five different honey species, all from Romania, overall 50 samples of honey. They measured important properties such as pH (3.88–6.39), aw (0.476–0.603), free acidity (3.40–37.10), MC (14.44–19.80%), EC (109.9–1276.8 µS/cm), ash content (0.05–0.63%), fructose content (33.64–47.31%), glucose (22.06–38.25%), sucrose (0–2.71%), as well as fructose and glucose contents sum (66.62–79.94%), etc. Based on LDA, 94% of the samples were correctly classified. Only the first two principal components (PCs) were important. The samples were grouped in five clusters based on their species, thus by botanical origin. However, in the meantime, the physico-chemical characteristics that may change during storage were also discussed. Regarding PCA, conductivity and ash content were found to be very important factors for the clustering of samples, however, free acidity and hue angle were not, due to similar species’ origins. The authors supported that pH is a factor influencing the extraction and storage of honey, as it affects stability, texture, and shelf life of honey. In addition, more than 20% moisture can speed up the fermentation reactions during storage [172].

Olawode et al. (2018) analyzed 10 honey samples based on their pH (3.75–4.38), EC (99–659 µS/cm), and moisture (14.2–17.7%) and their measurements agreed with quality limits. In addition, ^1^H-NMR profiling was measured, and after that PCA and PLS-DA were applied. The aim of the study was to classify the samples based on botanical or geographical origins; however, they also discussed processing and storage. Concerning ^1^H-NMR profiling, the peaks at 8.46 ppm (formic acid) and 9.49 ppm (HMF) were important in all honey samples. All the honeys contained HMF. The authors confirmed that the HMF in honey increases due to storage and thermal treatment of honey during processing [29]. D-glucose, D-fructose, maltose, and sucrose profiles were similar for all the honey samples and did not seem to be important in chemometric analysis. Normalization of the data using Log 2 transformation and Pareto scaling were used to make the metabolites’ concentration reasonably normal and more comparable. OPLS-DA supervised chemometric method was also applied, as it is a powerful tool for data reduction and identification of the most important spectral points for discrimination of samples. The authors stated that the OPLS-DA models produced were less complex and more meaningful than PLS-DA. OPLS-DA may replace PLS-DA, due to its ability to discriminate between variations in the normalized data that are important for predicting grouping. HCA took place using the centroid option for the observations, and Euclidean distances were calculated. The HCA dendrogram and the PLS-DA score plot were both in agreement [29].

Zhao et al. (2018) managed to discriminate overheated honey (industrial treatment) based on the analysis of amino acid and 5-HMF contents as well as color values after different thermal treatments. Two categories of honey were used, such as a light-colored honey (chaste honey) and a dark-colored honey (jujube honey). The authors found out that the concentrations of most amino acids in honey decreased after heat treatment, and also 5-HMF and proline in jujube honey, as well as 5-HMF and phenylalanine in chaste honey [257]. HCA, PCA, and OPLS-DA chemometric methods were applied to study the similarities and differences of the samples. More particularly, HCA and PCA, so the non-supervised methods were applied to the data with data mean-centered, UV-scaled, and log-transformed. A cluster tree was produced after application of HCA by using the group distance method, thus the distance represents the degree of similarity, based on heating time and heating temperature of honey samples. OPLS-DA was performed to discriminate the samples, and mean-centered, Pareto scaled, and log-transformed data were used. The seven-fold internal cross-validation was used to validate the OPLS-DA models, as well as permutation tests (20 times). Furthermore, their study may be used for identification of overheated (ultra-high temperature) processed honey (65 °C for 10 h or 80 °C for 8 h) versus moderate thermal conditions [257].

Ismail et al. (2021) used a variety of physico-chemical properties as well as characterization based on ATR-FTIR measurements to distinguish Malaysian honey samples from different species, dehumidification process, and geographical origins by applying chemometrics. Dehumidified honey samples were treated by dehydration using a dehumidifier at a temperature between 35 °C and 38 °C. Their study was innovative since no study has investigated the effect of dehumidification on the proline level in honey. The mean concentration of proline in dehumidified honey samples (14.97 mg kg^−1^) was significantly higher than in raw honey samples (5.52 mg kg^−1^). The conclusion was that physico-chemical properties, ATR-FTIR, and chemometrics are capable of differentiating honey samples according to the dehumidification process and geographical origin but not by species [199].

Regarding chemometric analysis, PCA was used and successfully grouped raw and dehumidified honey using both physico-chemical properties and FTIR data. The data were centered to 0 and scaled using a unit variance. For the visualization of results various techniques were applied, such as 2-D and 3-D score plots, biplot (overlay between 2-D score plots and loading plots), and eigenvalues plots. The statistical difference was tested using the unpaired *t*-test with Welch’s correction for normally distributed data, Mann–Whitney *U* test, and Spearman’s r test for parameters with non-normal distribution. *p*-values of less than 0.05 and 0.01 were considered significant [199].

Some honeys found in the local Malaysian market were dehumidified, and this trend is expected to increase in the future, as the authors stated; their goal was to observe the differences between raw and dehumidified honey. Twenty-five samples of dehumidified honey were compared with 49 samples of raw honey of the same species. The dehumidified group was found to have significantly lower water content (WC), fructose, and sucrose, however the group possessed significantly higher electrical conductivity (EC), insoluble matter (IM) content, and proline, as presented in Figure 2A. The parameters pH, free acidity (FA), glucose, maltose, HMF, and ash content (AC) were similar between raw and dehumidified samples. A clear separation was observed in the PCA biplot with the raw samples at the left side of the quadrant while dehumidified samples at the right side, as shown in Figure 2B.

Chemometric interpretation of the ATR-FTIR measurements showed that there were different wavenumbers in the raw honey when compared with the dehumidified stingless bee honey, as shown in Figure 3. The wavenumbers at 3242 cm^−1^ due to OH stretching, 2934 cm^−1^ is related to -CH stretching of carboxylic acids, 1657 cm^−1^ because of OH deformation, 1256 and 1040 cm^−1^ corresponding to the C-O stretch in the COH group and the C-C stretch in the carbohydrate structure were significantly present in raw samples. On the other hand, the wavenumbers at 700–978 cm^−1^ were prominent in dehumidified samples representing the out-of-plane OH deformation, C=O in-plane deformation, and vibrations of the CH_2_ group of L-proline.

Antonova et al. (2021) worked with honey samples to determine adulterations or changes caused due to thermal treatment. They stated that chemometrics had a significant role for the ATR-FTIR data interpretation. Three species of raw honey before and after thermal treatment, for various exposure periods and different temperatures have been tested. Calibration and validation models produced by chemometric analysis showed that the most useful region was 800–1500 cm^−^^1^ which contained characteristic bands of sugar transformations. Cross-validation took place based on the train-test-split approach, which randomly splits the data into a training set and a test set containing 75 and 25% of the data, respectively [215]. PCA successfully distinguished the samples between manually thermally treated and raw honey. It was used to initially decrease the huge load of data, so as to produce a new, smaller set of variables. PCA is not suitable for quantitative studies, but it is very useful for a general overview of the samples. The authors found that among the three species, eucalyptus honey changed the most after thermal treatment. As shown in Figure 4B, thermal processing caused significant changes in the ratio of the intensities of the bands at 990 cm^−^^1^ and 1050 cm^−^^1^, which represent fructose and glucose, respectively, and the reason could be either the Maillard reaction or sugar changes. As seen in Figure 4A, thermal treatment for 15 and 120 min at 70 °C cannot be separated very well by PCA [215].

On the other hand, LDA quantitatively discriminated against different conditions of thermal treatment at 70 °C, as presented in Figure 5 left. A confusion matrix was calculated for validation. In addition, heating at 40 °C at different periods is seen in Figure 5 right. Of course, the lower the heating temperature, the lower the classification that was possible, but raw honey was still clearly discriminated from heated honey. LDA also proved that the most important spectral wavenumber for differentiation is approximately at 990 cm^−^^1^. Antonova et al. (2021) stated that ATR-FTIR spectroscopy with chemometric methods proves a powerful technique as they can detect if heating was due to transport or storage or the intended adulteration of raw honey [221].

Chen et al. (2014) have studied manually dehydrated raw honey by using synchronous two-dimensional (2D) NIR correlation spectroscopy. Honey samples were taken from six different dehydration stages using a drum wind-drying method with the temperature monitored at 40 °C. The second overtone of O–H and N–H groups vibration upon their H-bonds forming or collapsing due to the interactions between water and solute. Problems such as baseline shift and overlapping peaks of raw spectra were solved by performing preprocessing using a chemometric software. The 25-point Savitzky–Golay (SG) quadratic polynomial smoothing was used, as well as 25-point quadratic polynomial first derivative to reduce baseline shift and enhance the spectral features. The synchronous 2D correlation contour maps between 900–1080 nm were obtained from the first derivative spectra of chaste honey at different drying stages. The authors concluded that absorption in the NIR short wave region was much weaker than that in the middle and long wave region (1100–2500 nm), and it was also overshaded by water absorption. Only the use of advanced chemometrics can handle that problem as the application enlarges the signal by minimizing the noise [248].

Yan et al. (2022) investigated a unique marker (327.1321 Da) that is produced after thermal heating (dehydration) by using UHPLC-Q-TOF-MS, HRMS, and NMR. N-(1-deoxy-1-fructosyl) phenylalanine (Fru-Phe) was identified, and it was an Amadori compound. The concentration of Fru-Phe was almost stable in naturally heated samples for more than 2 years of storage, but it increased in manually heated samples, and this marker can indicate honey fraud. The PCA method was used to visualize the groups of samples and to check that the groups were well-separated [262].

Rust et al. (2021) used NIR spectroscopy combined with chemometric analysis for determining syrup-adulterated honey. The effects of age, storage temperature, syrup adulteration (10 and 20% *w*/*w*), and irradiation treatment were captured, and ANOVA-simultaneous component analysis (ASCA) was used to treat the data which is a method combining ANOVA and PCA. Pre-processing by standard normal variate (SNV) was performed to eliminate unwanted multiplicative effects from the spectra. The proposed method was successful to detect syrup-adulterated honey [219].

Pasias et al. (2022) tried to determine the optimum conditions for honey storage. To obtain a non-crystallized product that lasts over a year, samples must be treated by heating at 72 °C or stored at −18 °C in order to maintain the same quality (low HMF content, high diastase activity, and high phenolic content) as with the fresh non-heated samples. PCA and HCA chemometric methods were applied to cluster the samples, and to distinguish and confirm the outcomes of the study [240]. Önür et al. (2018) assessed three methods of processing in honey, such as thermal, ultrasound (US), and high hydrostatic pressure (HHP) processing on the liquefaction of honey. US generated rapid dissolution of crystals. HHP gave shorter liquefaction times, as well as relatively lower HMF formation observed with HHP treatments. Chemometric analysis took place via the PCA method, and the contribution was in explaining the importance and the relation among various factors especially in the case of ultrasonication variables which were interrelated [112]. Scripcă and Amariei (2021) also used PCA to study the correlations between the honey types and the sugars, color parameters, and texture parameters. The PCA method was also important to see the variation and to finalize the main principle components [115].

## 5. Conclusions

Dewatering/dehumidification and thermal treatments, together with storage, are the most important stages of honey processing. They influence honey composition such as the concentration of sugars, organic acids, amino acids, and the content of HMF and phenolics, physico-chemical characteristics such as the free acidity and diastase activity, water content or processes such as the honey crystallization and melting. Honey composition and the physico-chemical parameters undergo significant changes through chemical and enzymatic reactions that take place and subsequently the changes in chemical and physical structures may affect its quality. 

Chemometric analysis is a useful and necessary approach, as well as an alternative way of handling experimental data. Validation increases the robustness of chemometric models to underline the quality of chemometric studies. Without using chemometrics, the results of the studies would not be possible to be drawn, thus the use of chemometrics seems crucial. Between 2014 and 2022, the growing number of research papers which use chemometrics in analyzing honey samples shows its current importance and effectiveness in the honey industry. The main reasons justifying the increasing demand for chemometrics are initially to interpret the huge quantities of measurements obtained by the analytical methods, and after this, the visualization of samples in groups for the unsupervised methods. Moreover, chemometrics are used to clarify the reason for grouping by identifying a marker responsible for grouping the samples. In addition, it enables prediction for a possible classification of unknown samples for the supervised methods. The research potential of chemometrics seems to be very positive. The application of chemometrics on honey composition/physico-chemical parameters alternations has become a valuable tool for revealing trends and patterns hidden in data based on which processing/storage affect honey quality. Chemometrics enable the explanation and understanding of the meaning of the numbers, in order to optimize quality control and safety protection leading to a better service to society, both to the industry performance and to consumers by ensuring high quality products.

## Figures and Tables

**Figure 1 foods-12-00473-f001:**
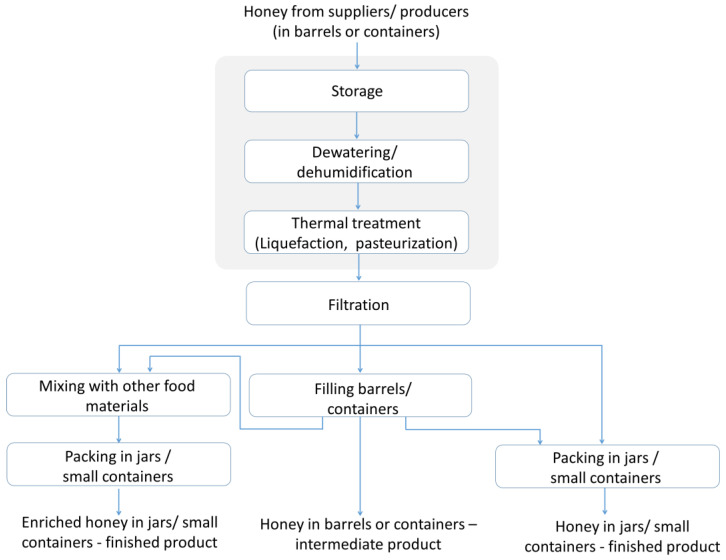
Steps in processing honey from suppliers or producers, highlighting the main operations where modifications can occur.

**Figure 2 foods-12-00473-f002:**
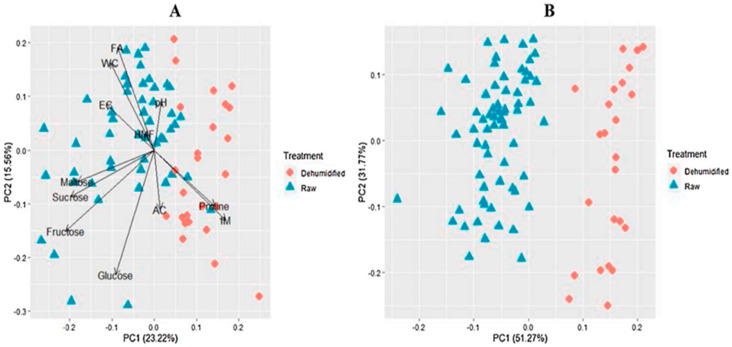
(**A**) PCA loading plot and (**B**) PCA score plot. Adopted from [196].

**Figure 3 foods-12-00473-f003:**
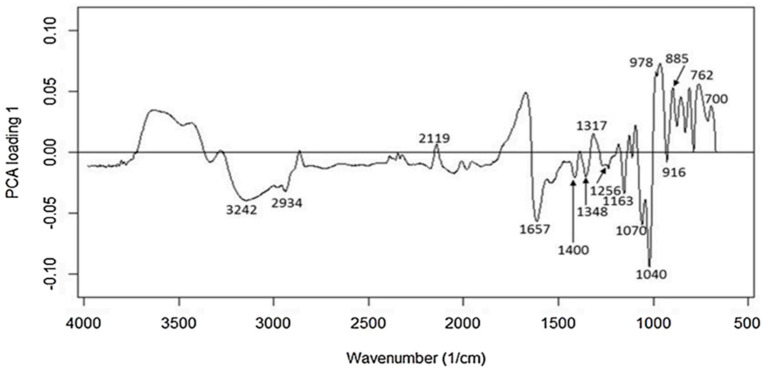
ATR–FTIR spectrum. Adopted from [193].

**Figure 4 foods-12-00473-f004:**
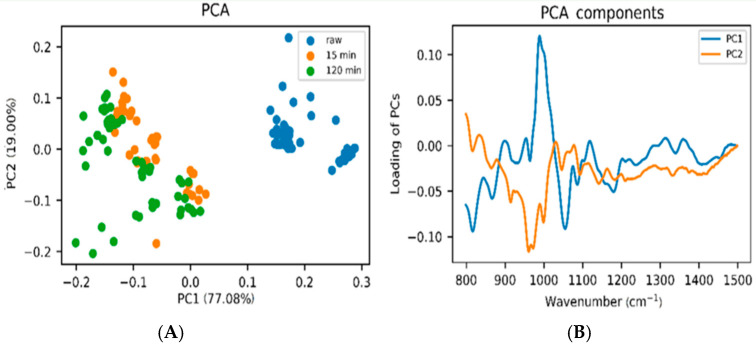
Based on ATR-FTIR data (**A**) PCA score plot and (**B**) loading plot for raw and thermally treated eucalyptus honey samples. Adopted from [215].

**Figure 5 foods-12-00473-f005:**
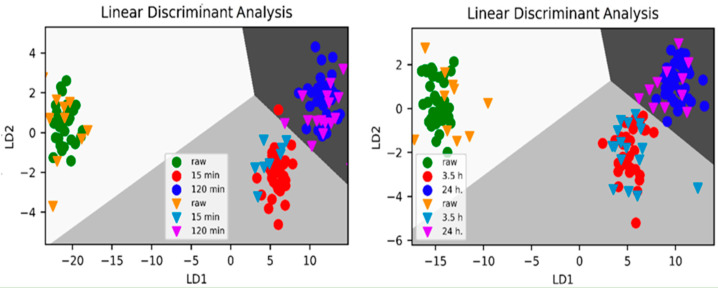
LDA plots based on ATR-FTIR data of the region 800–1500 cm^−^^1^; left—Samples heated at 70 °C, right—Samples heated at 40 °C. Dots: training set, triangles: test set. Adopted from [215].

**Table 1 foods-12-00473-t001:** Honey dewatering and dehumidification methods used in the last 12 years.

Method	Device and Conditions	Reference
Use of dry air	Heating in a desiccant honey dehydrator (with silica gel desiccant bed) with dehumidified air at 35 °C or 45 °C	[99]
	Heating of containers (having hot-water jacket) combined with treatment with dried air (until 40 °C)	[100]
	Desiccant-bed silica gel heating and drying the air, with recirculation at 40–55 °C up to 36 h	[101]
	Dehydrator system with control of temperature, drying air speed, relative humidity and honey exposure surface	[102]
	Vertical centrifugal honey-dehydrator with an external electric heat source and a closed air circuit and heat pump	[103]
Vacuum drying	Use of Low Temperature Vacuum Drying (LTVD) (30 °C) with induced nucleation technique	[104]
	Ultrasonic vacuum drying at 40 kHz 80 W	[105]

**Table 2 foods-12-00473-t002:** Chemometrics used in the honey quality analysis.

Chemometrics	Group Known a Priori	Independent Variables	Dependent Variable	Source Data	Output	Reference
ANOVA MANOVA	no	Categorical	Continuous	IPC-MS	Geographical origin	[139]
				(HS-SPME/GC-MS)	Significant VOC for geographical origin	[145]
PARAFAC		Categorical	Continuous	Fluorescence spectrometry data	Adulterants	[134]
CA	no			DLLME-GC-MS	Geographic grouping of pollutants in honey	[128]
				DLLME-GC-MS	Anthropic grouping of pollutants in honey	[129]
				IPC-MS	Geographical origin	[139]
				GC-MS	Relevant physico-chemical parameters and VOC	[169]
PCA	no	Continuous	Continuous	DLLME-GC-MS	Geographic grouping of pollutants in honey	[128]
				Vitamin B2 and Cu, antioxidant activity	Honey type	[131]
				HPLC data + ATLD	Honey type	[138]
				IPC-MS	Geographical origin	[139]
				(HS-SPME/GC-MS)	Significant groups of geographical origin	[145]
				SWATH-MS	Honey adulteration, geographical and botanical origin	[138]
				GC-MS	Relevant physico-chemical parameters and VOC	[169]
				Raman spectra tSNE	botanical origin and the quantity of adulterants	[177]
LDA	yes	Continuous	Categorical	IPC-MS	Geographical origin	[139]
				FTIR spectra with pre-processing	Botanical origin	[143]
				(HS-SPME/GC-MS)	Geographical origin	[145]
				Fluorescence spectrometry data	Adulterants	[134]
				PCA of MALDI-ToF-MS	Botanical origin	[162]
				PCA of MIR spectra	Botanical origin	[162]
				Gold nanoparticle sensor array	Botanical origin	[170]
				Physico-chemical and rheological parameters	Botanical origin	[173]
				PCA of Raman spectra	Quantity and type of multiple adulterants	[175]
PLS	yes	Continuous	Continuous	FTIR spectra with pre-processing	pH, electrical conductivity, free acidity, 5-HMF, fructose, glucose and sucrose	[143]
				Raman spectra	Botanical origin and the quantity of adulterants	[177]
PLS-DA	yes	Continuous	Categorical	^1^H NMR spectra with pre-processing	Geographical and botanical origin	[149]
				^1^H NMR spectra	Honey adulteration	[155]
				SWATH-MS	Honey adulteration, geographical and botanical origin	[138]
				Differential pulse voltammetry using NADES	Alteration after heat treatment	[158]
				ATR–FTIR spectra with pre-treatment	Adulteration and botanical origin	[166]
				GCMS	Biological origin	[169]
				Gold nanoparticle sensor array	Botanical origin	[170]
				Raman spectra	Quantity and type of multiple adulterants	[175]
kNN	yes	Categorical	Categorical	PCA of MALDI-ToF-MS	Botanical origin	[162]
				PCA of MIR spectra	Botanical origin	[162]
				Raman spectra	Quantity and type of multiple adulterants	[175]
SIMCA	yes	Continous	Categorical	PARAFAC of fluorescence spectra	Geographical and botanical origin	[161]
				MALDI-ToF-MS	Botanical origin	[162]
				MIR spectra	Botanical origin	[162]
SVM	yes	Continuous	Continuous	ATR–FTIR spectra with pre-treatment	Adulteration and botanical origin	[166]
				GCMS	Biological origin	[169]
				Gold nanoparticle sensor array	Botanical origin	[170]
				PCA and tSNE of Raman spectra	Botanical origin and the quantity of adulterants	[175]
ANN	yes	Continuous	Continuous	Raman spectra	Maltose, fructose and sucrose content	[172]
				Physico-chemical and rheological parameters	Botanical origin	[173]
				Raman spectra	Botanical origin and the quantity of adulterants	[175]

**Abbreviations:** ANOVA = analysis of variance; MANOVA = multivariate analysis of variance; PARAFAC = parallel factor analysis; HCA = hierarchical cluster analysis; CA = cluster analysis; LDA = linear discriminant analysis; PCA = principal component analysis; PLS = partial least square; PLS-DA = partial least square-discriminant analysis; kNN = k–nearest neighbor; SIMCA = soft independent modeling of class analogy; SVM = support vector machine; ANN = artificial neural network; IPC-MS = inductively coupled plasma-mass spectrometry; HS-SPME/GC-MS = headspace-solid phase microextraction-gas chromatography-mass spectrometry; DLLME-GC-MS = dispersive liquid-liquid microextraction-gas chromatography mass-spectrometry; GCMS = gas chromatography mass spectrometry; HPLC data + ATLD = high-performance liquid chromatography data + alternating trilinear decomposition; SWATH-MS = Sequential Window Acquisition of All Theoretical Mass Spectra; tSNE = t-distributed stochastic neighbor embedding (algorithm); MALDI-ToF-MS = matrix-assisted laser desorption ionization time-of-flight mass spectrometry; MIR = mid-infrared, FTIR = Fourier transform infrared; ^1^H NMR = proton nuclear magnetic resonance; NADES = natural deep eutectic solvents; ATR–FTIR = attenuated total reflection - Fourier transform infrared.

**Table 3 foods-12-00473-t003:** Research studies related to quality characteristics in honey during processing and storage.

Quality Characteristics Analyzed	Analytical Method(s)	Chemometric Method(s)	Reference
Water content, EC, AC, pH and FA, HMF, IM, proline, sugar profile (fructose, glucose, maltose and sucrose)	Digital refractometer, EC meter, electrical furnace, pH meter, spectrophotometer, HPLC-RID and ATR-FTIR	PCA	[199]
Free amino acids, color and 5-HMF	HPLC-DAD and colorimeter	PCA, HCA and OPLS-DA	[257]
Water content, pH, sugar content (glucose, fructose, and sucrose) and HMF	Refractometry, pH meter, GC-FID, UHPLC-PAD and ATR-FTIR	PCA and LDA	[221]
EC, water content, pH and HMF	pH meter, EC meter and ^1^H-NMR	PCA, PLS-DA, OPLS-DA and HCA	[29]
Glucose, fructose and sucrose content, pH, water content, a_w_, refraction index, Brix concentration, FA, ash content, EC and color parameters	Chromatography, pH meter, refractometer, conductivity meter and colorimeter	PCA and LDA	[173]
Water content, electric conductivity, pH, free acidity and lactones, diastase index, UV/Vis spectrophotometer, color and NIR measurements	Refractometer, conductivity meter, pH meter, UV/Vis spectrophotometer, UHPLC-RID, visible spectrophotometer and NIR spectrometer	PCA, PLS-DA and SVM	[247]
Reduction of water content and volatile components by evaluating H-bonds forming or collapsing in the vibrations of H-bonded groups due to thermal hydration	NIR	Synchronous 2D correlation analysis	[248]
N-(1-deoxy-1-fructosyl) phenylalanine (Fru-Phe), an Amadori compound which is produced in the first stages of the Maillard reaction due to thermal hydration	UHPLC-Q-TOF-MS, HR-MS and NMR	PCA	[262]
Spectral regions related to age, temperature, and syrup adulteration of honey	NIR	ASCA	[219]
HMF, diastase activity and phenolic content	UV–visible spectrometry and chromatography	PCA and HCA	[240]
Physico-chemical properties (liquefaction time, diastase number, color and viscosity and HMF formation)	Rheometer, spectrophotometer, HPLC	PCA	[112]
Water content, acidity, water activity, glucose, fructose, sucrose, glucose/water ratio, glucose/fructose ratio, textural parameters (hardness, springiness, cohesiveness, adhesiveness, viscosity, chewiness and gumminess), microbial number and content of crystals	Refractometer, HPLC, UV-VIS spectrophotometer, colorimeter, texture analyzer, stereomicroscope	PCA	[115]

**Abbreviations:** ASCA = ANOVA-simultaneous component analysis, a_w_ = water activity, AC = ash content, ATR-FTIR = Attenuated total reflection Fourier–transform infrared, DAD = diode array detector, EC = electrical conductivity, FA = free acidity, GC-FID = gas chromatography with flame ionization detector, HCA = hierarchical cluster analysis, 5-HMF = 5-hydroximethylfurfural, HPLC = high-performance liquid chromatography, HPLC-DAD = high-performance liquid chromatography with diode-array detection, HPLC-RID = high-performance liquid chromatography with a refractive index detector, HR-MS = high resolution-mass spectrometry, IM = insoluble matter, LDA = linear discriminant analysis, M = moisture content, NIR = near infrared, NMR = nuclear magnetic resonance, PCA = principal component analysis, PLS = partial least square, PLS-DA = partial least square—discriminant analysis, OPLS-DA = orthogonal projections to latent structures discriminant analysis, SVM = support vector machine, UHPLC-PAD = ultra-high-performance liquid chromatography equipped with a photodiode-array detector, UHPLC-Q-TOF-MS = ultra-high-performance liquid chromatography equipped with Quadrupole-time of flight and mass spectrometer, UHPLC-RID = ultra-high-performance liquid chromatography equipped with refractive Index Detector.

## Data Availability

Not applicable.

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
