# Peer review of "Insight into the Recent Application of Chemometrics in Quality Analysis and Characterization of Bee Honey during Processing and Storage"

_foods, 2023, doi:10.3390/foods12030473_

Round 1
Reviewer 1 Report
I find this manuscript an impressive collection of information. However, it has a seriously missing piece. Bruker has a highly professional, industrialized protocol for honey analysis (link below), which should not be ignored.In general, NMR spectroscopy (not only 1H NMR) is not discussed at its level of importance, in my view.
Reviewer 2 Report
see attached PDF

Reviewer 3 Report
The "aim of the study" is a current topic and read-worthy. However, this manuscript on the use of chemometry in honey characterization and determination of quality parameters was not as explanatory as expected. However, plagmarism is quite a lot.
For this reason, it has been evaluated as a major revision. It is necessary that these serious similarities are edited and expressed more clearly in the terms "honey and chemometry".
Round 2
Reviewer 3 Report
The authors made the requested changes, satisfied the reviewer.